# Computing Preimages of Deep Neural Networks with Applications to Safety

## Abstract

To apply an algorithm in a sensitive domain it is important to understand the set of input values that result in specific decisions. Deep neural networks suffer from an inherent instability that makes this difficult: different outputs can arise from very similar inputs.

We present a method to check that the decisions of a deep neural network are as intended by constructing the exact, analytical preimage of its predictions. Preimages generalize verification in the sense that they can be used to verify a wide class of properties, and answer much richer questions besides. We examine the functioning and failures of neural networks used in robotics, including an aircraft collision avoidance system, related to sequential decision making and extrapolation.

Our method iterates backwards through the layers of piecewise linear deep neural networks. Uniquely, we compute *all* intermediate values that correspond to a prediction, propagating this calculation through layers using analytical formulae for layer preimages.

## 1 Introduction

Folk wisdom holds that although deep neural networks (DNNs) can achieve excellent predictive accuracy, reasoning about their performance is difficult, even for experts. Our goal is to enable non-expert stakeholders, such as clinical health workers, investors, or military commanders to build trust a statistical model in high-stakes environments. To do this, we posit that decisionmakers want to understand a model in both directions, both from inputs to outputs, but also being able to start with hypothetical outputs, and understand the inputs that lead to them.

In this paper, we develop an equivalent, but much simpler, representation of a certain class of DNN classifiers. This representation, which requires only a basic numeracy to productively interact with, can be used by domain experts to build intuition and trust. We apply this method to a reinforcement learning agent trained to solve the cart-pole problem, and find that a DNN implementing a successful policy makes a particular type of mistake on $24\%$ of the mass of the $1/8$th of the state space for which we know the optimal action (Section 3.2). We also show how using the preimage in place of verification can yield a more efficient and interpretable end-to-end system for analyzing aircraft collision avoidance systems (Section 3.3).

### 1.1 Previous work

DNNs have the property that knowing the output tells us very little about the input it corresponds to. This is most apparent in image classifiers, where totally different outputs can arise from inputs that are visually indistinguishable (Szegedy et al. (2014)). We build upon the mathematical framework developed for verifying DNNs that grew out of a desire to prove the absence of adversarial examples, for example Tjeng et al. (2017) and Wong & Kolter (2017). However, we depart from these studies along with Katz et al. (2017), being more oriented towards small DNNs that map to and from low-dimensional spaces with considerable structure. These DNNs arise especially in systems which interoperate with the physical world, for example mapping measurements of positions and velocities to movements. Table 1 orients our work to the literature.

| Commonly called | What is computed | Examples |
|---|---|---|
| Verification | $(f, X, Y) \mapsto \mathbf{1}_{f^{-1}(Y) \cap X = \emptyset}(= \mathbf{1}_{f(X) \cap Y = \emptyset})$ | Wong & Kolter (2017) |
| Reachability | $(f, X) \mapsto f(X)$ | Yang et al. (2020) |
| Inversion | $(f, y) \mapsto f^{-1}(\{y\})$ | Carlsson et al. (2017) |
| Preimage | $(f, Y) \mapsto f^{-1}(Y)$ | This paper |

Table 1: A taxonomy of previous work on inversion and verification. Here $f : \mathbb{R}^{n_1} \to \mathbb{R}^{n_L}$ is a DNN, $X \subseteq \mathbb{R}^{n_1}$, $x \in \mathbb{R}^{n_1}$, $Y \subseteq \mathbb{R}^{n_L}$, and $y \in \mathbb{R}^{n_L}$. $f^{-1}$ is its inverse in the sense that $f^{-1}(Y) = \{x : f(x) \in Y\}$.

We have phrased verification in this unusual fashion to facilitate comparison with the other points. Stated in the familiar application to image classifiers $X$ would be an epsilon ball around an input, and $Y$ would be the halfspace where one coordinate is higher than all others.

Verification ultimately amounts to a simple yes or no, and so answering higher-level questions typically requires many verifications: for example, Katz et al. (2017) describes a suite of 45 tests, and image classifiers often wish to verify the absence of adversarial examples around the entire training set. Yang et al. (2020) is an interesting extension to verification in that it computes the entire image of, say, an epsilon ball around a data point, and not just whether it intersects with a decision boundary.

Reasoning forward, about the outputs that can arise from inputs, is only half of the picture. Carlsson et al. (2017) and Behrmann et al. (2018) are oriented backwards, they attempt to reconstruct the inputs that result in an output. These related papers study the statistical invariances that nonlinear layers encode. Behrmann et al. (2018) examines the preimage of a single point through a single ReLU layer, analyzing stability via an approximation-based experiment. Carlsson et al. (2017) analyzes the preimage of a single point through the repeated application of a nonlinearity, purely theoretically. Our paper looks at the preimage of non-singleton subsets of the codomain, which is much more practically useful, and requires considerable extension to their approaches.

## 2 METHOD

Our method is easily stated: build up the preimage of a DNN from the preimage of its layers, using simple analytical formulae. We start by developing some properties of the preimage operator, then we describe the class of sets that we compute the preimage of, and finally we discuss the class of DNNs that our algorithm addresses.

### 2.1 PROPERTIES OF PREIMAGES

Lemma 1 shows how to build up the preimage of a DNN from the preimages of its constitutent layers.

**Lemma 1** (Preimage of composition is reversed composition of preimages). *For functions* $f_j : \mathbb{R}^{n_j} \to \mathbb{R}^{n_{j+1}}$,

$$(f_{\ell+k} \circ f_{\ell+k-1} \circ \ldots \circ f_\ell)^{-1} = f_\ell^{-1} \circ \ldots \circ f_{\ell+k-1}^{-1} \circ f_{\ell+k}^{-1}. \tag{1}$$

Secondly, we mention an intuitive property of $f^{-1}$ that is handy for building up the preimage of any set from the preimages of any partition of that set.

**Lemma 2** (Preimage of union is union of preimages).

$$f^{-1}\left(\cup_{i=1}^{N} S_i\right) = \cup_{i=1}^{N} f^{-1}(S_i).$$

### 2.2 POLYTOPES

Our method is not applicable to arbitrary sets $Y$, but rather sets that, roughly, have piecewise linear boundaries. The basic building block of these sets are polytopes.

**Definition 1** (Polytope). *A polytope in $\mathbb{R}^n$ is a set that can be written as $\{x \in \mathbb{R}^n : b - Ax \geq 0\}$ for some $m \in \mathbb{N}$, $b \in \mathbb{R}^m$, and $A \in \mathbb{R}^{m \times n}$.*

Put more simply: a polytope is the intersection of half-planes. Definition 1 does not require that polytopes be bounded, but polytopes are convex. Sets with linear boundaries, though they may be non-convex, can decomposed into the union of polytopes. We term such sets *region-unions*, and the set of polytopes which comprise them, *regions*.

**Definition 2** (Region and region-union). *For $N \in \mathbb{N}, b_i \in \mathbb{R}^{m_i}$, $A_i \in \mathbb{R}^{m_i \times n}$, with $m_i \in \mathbb{N}$, a region is*

$$\{\{x : b_i - A_i x \geq 0\} ; i = 1, \ldots, N\} . \tag{2}$$

*A region-union is a set $\cup_{r \in R} r$ for some region $R$.*

Region-unions are interesting because the the preimage polytopes under piecewise linear functions are regions-unions. However, we need to also keep information on how to form a region-union, hence the notion of a region. It is trivial to observe that if $R_1$ and $R_2$ are regions, then $R_1 \cup R_2$ is likewise a region, and correspondingly for region-unions.

## 2.3 LINEAR AND RELU POLYTOPE PREIMAGES

In this section, we give formulae for the preimage of linear and ReLU functions, giving significant content to Lemma 1. The preimage of polytopes under linear mappings are polytopes:

**Lemma 3** (Preimage of Linear layer).

$$(x \mapsto Wx + a)^{-1}(\{x : b - Ax \geq 0\}) = \{x : (b - Aa) - AWx \geq 0\}. \tag{3}$$

ReLU is a piecewise linear function, so if we carefully treat the portions of the domain on which it exhibits different behavior, we obtain a similar formulation for each:

**Lemma 4** (Preimage of ReLU layer).

$$\begin{aligned} &\text{ReLU}^{-1}(\{x : b - Ax \geq 0\}) \\ &= \bigcup_{\nu \in \{0,1\}^n} \{x : b - A\text{diag}(\nu)x \geq 0, -\text{diag}(1 - \nu)x \geq 0, \text{diag}(\nu)x \geq 0\} . \end{aligned} \tag{4}$$

To understand Lemma 4 let $s(x)$ be the vector given by $s(x)_i = 1$ if $x_i \geq 0$ and zero otherwise. Then $\text{diag}(s(x))x = \text{ReLU}(x)$. This expression separates $x \mapsto \text{ReLU}(x)$ into a pattern of signs over its coordinates and $x$ itself. This means that once we restrict attention to a set on which the sign does not change, we can apply familiar linear algebra routines to compute the preimage set, akin to Lemma 3. The nonnegative values are denoted by $\nu \in \{0,1\}^n$ in the above, and the set of $x$ such that $x_i \geq 0 \iff \nu_i = 1$ is given by $\text{diag}(\nu)x \geq 0$. Similarly, $x_i \leq 0 \iff \nu_i = 0$ for $i = 1, 2, \ldots, n$ if and only if $-\text{diag}(1 - \nu)x \geq 0$. Equation 4 follows by partitioning $\mathbb{R}^n$ into the $2^n$ sets where each coordinate is nonnegative or not.

Computing the preimage of a ReLU layer is unavoidably intractable at scale, though the problem exhibits considerable structure. We expect that it is possible to compute the preimage of networks of a similar scale to those that can be completely verified, such as small image-scale networks. Preimages are most insightful and useful when the inputs and outputs have definite interpretation – application areas where the need for massive networks is less.

## 2.4 THE SUFFICIENCY OF LINEAR AND RELU LAYERS

In familiar terms a DNN classifier might consist of some "feature building" modules, say composed of alternating convolution and maxpooling, then flattened, and passed onto the prediction logic consisting of alternating linear and ReLU layers, possibly including dropout or batch normalization, and concluding with a softmax function to normalize the predictions to a probability distribution. Resnets (He et al. (2016)) do not strictly fit this pattern, bu can be handled with similar reasoning (see Appendix B).

How do the results of Section 2.3 suffice to invert such DNNs? Firstly, under our convention that layers operate on flat tensors, flattening is superfluous. Next, dropout affects inference only through the weights – this layer can be omitted entirely in computing the preimage. Convolution is essentially linear. Maxpool is straightforwardly rewritten in terms of the ReLU and linear function. $\{x : b - A\text{softmax}(x) \geq 0\}$ is not a polytope. However, if the classification alone suffices then the softmax layer can be elided entirely since $\arg\max_j x_j = \arg\max_j \text{softmax}(x)_j$.

# 3 EXPERIMENTS

## 3.1 TWO MOONS CLASSIFICATION

To cultivate some intuition about the preimage of a DNN we start by examining a classic test problem in nonlinear classification. We fit a DNN $f : [-3, +3]^2 \to \mathbb{R}^2$ consisting of two nonlinear layers with eight neurons each on an instance of the "two moons" dataset. This data is shown in Figure 1a (further details of details of $f$ and the data are in Section D.1). Figure 1b plot the corresponding logits, along with the sets to be inverted $\{x : x_1 \lessgtr x_2\} \subseteq \mathbb{R}^2$. Figure 1c shows the corresponding preimages, with different hues of the same color corresponding to different sign patterns $\nu$ in Equation 4.

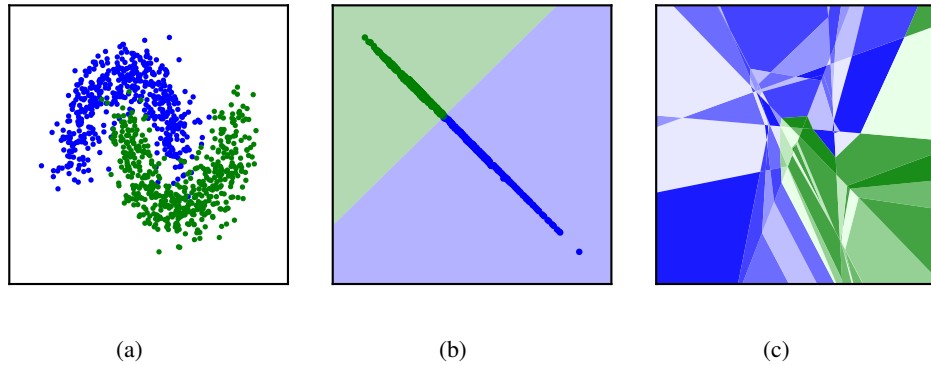

(a)                              (b)                              (c)

Figure 1: Inversion of a simple DNN $\mathbb{R}^2 \mapsto \mathbb{R}^2$ fit to the "two moons" data shown in Figure 1a. In Figure 1b are the logits from a simple DNN corresponding to each data point computed with an inference pass, along with the decision boundary. Figure 1c shows the preimages comprised of polytopes that form the region-union.

## 3.2 CART-POLE REINFORCEMENT LEARNING AGENT

In the "cart pole" control problem a pole is balanced atop a cart which moves along a one dimensional track (Figure 2). Gravity pulls the pole downward, the falling of the pole pushes the cart, and external movement of the cart pushes the pole in turn. The control problem is to keep the pole upright by accelerating the cart.

In the formulation of Brockman et al. (2016) controller inputs are: the position of the cart, $x$, velocity of the cart $\dot{x}$, the angle of the pole $\theta$ from upright, and the angular velocity of the pole $\dot{\theta}$. Possible actions are to accelerate the cart in the positive or negative $x$ direction. The reward environment encourages balancing by a unit reward per period before failure, where failure means that the pole is not sufficiently upright ($\theta \notin [-\pi/15, +\pi/15]$), or the cart not near enough the origin ($x \notin [-2.4, +2.4]$). We have no prescribed limits for $\dot{x}$ and $\dot{\theta}$, but via a methodology described in Section D.2.1, we interpret these states as taking values in $[-3.0, +3.0] \times [-3.5, +3.5]$.

Consider a still cart and pole ($\dot{x} = \dot{\theta} = 0$), with the cart left of zero ($x \leq 0$) and the pole left of vertical ($\theta \leq 0$). Keeping $x$ and $\theta$ near zero is preferable, since these are further from failure, so moving left will steady $\theta$ but worsen $x$. Nonzero velocities make this reasoning more complicated, but one configuration is unambiguous: if $x \leq 0, \dot{x} \leq 0, \theta \geq 0, \dot{\theta} \geq 0$, then pushing right is clearly

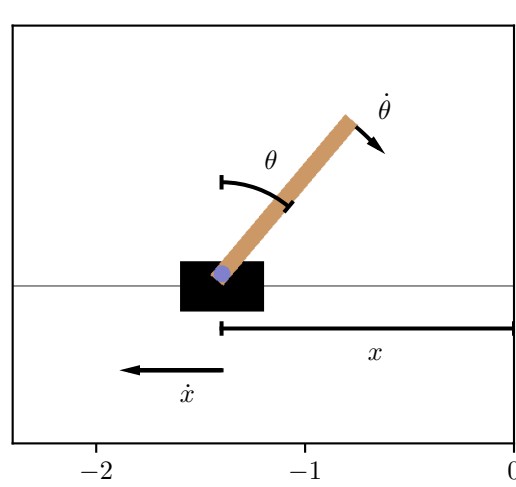

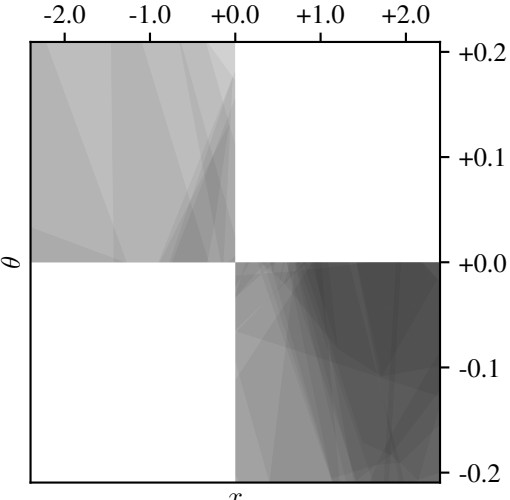

Figure 2: The state space of the cart pole problem, schematically. Here $x \leq 0$ (the cart is left of the origin), $\dot{x} \leq 0$ (the cart is moving leftward), $\theta \geq 0$ (the pole is right of vertical), and $\dot{\theta} \geq 0$ (the pole is moving rightward).

Figure 3: Projection of subsets of the domain where the wrong action is taken, with the hue of the area being proportional to the volume of the wrong sets, divided by the volume of the projection.

the correct action. Figure 2 gives depicts a value in this orthant. Let $D_{+1} = (-\infty, 0]^2 \times [0, \infty)^2$, and correspondingly, let $D_{-1} = [0, +\infty)^2 \times (-\infty, 0]^2$.

We fit a one hidden layer neural network control function $f : \mathbb{R}^4 \to \mathbb{R}^2$ using policy gradient reinforcement learning. Details of this calculation are in Section D.2. This agent achieves its goal of balancing the pole: in 1000 trials of 200 periods, $(x, \theta)$ remains uniformly in $[-.75, +.75] \times [-.05, +.05]$ with very low velocities. Nonetheless there are many states for which pushing right is clearly the correct action, but for which the DNN controller predicts $-1$: in the same simulation of 1000 trials of 200 steps, roughly 7% of actions performed by the agent fail this sanity check. This behavior is not a numerical fluke – it holds if we consider states only nonnegligibly interior to $D_{+1}$ and $D_{-1}$, and also if we only count predictions that are made with probability greater than .51. One such pockets of counterintuitive behavior is

$$[-2.399, -1.462] \times [-2.922, -2.262] \times [+1.798 \times 10^{-8}, +0.1067] \times [+1.399, +1.728] \subseteq$$
$$D_{+1} \cap f^{-1}(\{x \in \mathbb{R}^2 : x_1 > x_2\}).$$

We find this box large – for example the first coordinate comprises almost 20% of that dimension of the state space. The size of this box is even more suprising because it is inscribed within a larger polytope (using the algorithm of Bemporad et al. (2004)) that has a volume about 40 times larger. The total volume in $\mathbb{R}^4$ of these sets is 3% of the state space volume, and thus 24% of the volume of $D_{-1} \cup D_{+1}$. Figure 3 parses this surprising fact a bit further by plotting the projection of the four-dimensional domain onto the $(x, \theta)$ plane. The hue of the gray is proportional to the volume of the four-dimensional polytope divided by the volume of the two-dimensional projection, so darker areas mean more $(\dot{x}, \dot{\theta})$ mass that is wrong. Since the entirety of the second and fourth quadrants are grey at *every* $(x, \theta) \in [-2.4, +2.4] \times [-\pi/15, +\pi/15]$ there are some $(\dot{x}, \dot{\theta})$ where the wrong action will be taken.

## 3.3 COLLISION AVOIDANCE SYSTEMS

The final application shows how to use domain knowledge to anticipate dangerous behavior of a DNN in a complex modelling domain.

### 3.3.1 BACKGROUND

Aircraft automated collision avoidance systems (ACAS) are navigational aids that use data on aircraft positions and velocities to issue guidance on evasive actions to prevent collisions with an intruding aircraft. The ACAS developed in Kochenderfer & Chryssanthacopoulos (2011) uses dynamic programming to formulate the optimal control of a partially observed Markov process, and issues advisories to optimize a criterion that penalizes near collisions and raising false or inconsistent warnings. Unfortunately, evaluating the policy function is too resource-intensive to run on certified avionics hardware. Small DNNs have been been found to be adequate approximators that require little storage and can perform inference quickly. A downside of this approximation is that even accurate DNNs can give very wrong predictions on some inputs – Katz et al. (2017), for example show that when another aircraft is nearby and approaching from the left, a DNN-based approximation need not advise the correct action of turning right aggressively.

Verification can check that one-step behavior in a DNN-based ACAS behaves as intended. However, it cannot answer higher level questions like "will a near-collision occur if this policy is followed?" The idea of Julian & Kochenderfer (2019) is to verify dynamic properties of such systems by combining single-step verification with worst-case assumptions about randomness in state transitions and (constrained) behavior of other aircraft.

### 3.3.2 DISCRETIZE AND VERIFY: JULIAN & KOCHENDERFER (2019)

In Julian & Kochenderfer (2019), the state consists of $x$ and $y$ distances between the two aircraft, and an angle of approach angle between them, $\psi$. The actions are five turning advisories: (1) "clear of conflict" (COC), (2) weak left [turn] (WL), (3) strong left (SL), (4) weak right (WR), and (5) strong right (SR). The initial condition is given by the boundary of the domain where the distance of the intruding aircraft are at their maxima. Transition dynamics are denoted by $\Psi(a, S)$, a set-valued function which gives the set of states that are reachable from states in $S$ under action $a$. $\Psi$ encompasses both randomness in the transition, and behavior of the other aircraft. The change in $(x, y)$ is controlled by the angle between the crafts, and the update to the angle is the difference between the turning of the two crafts, with some randomness. To compute the states that can arise under a policy, the idea is to begin from an initial set of states that are known to be reachable, and to iteratively append states that are reachable from any of those states, until a fixed point is reached. $U$ denotes the set of states that we wish to preclude.

| $g$ | Volume fraction |
|-----|-----------------|
| 40  | 0.05128 |
| 80  | 0.02532 |
| 120 | 0.01681 |
| 160 | 0.01267 |
| 200 | 0.01005 |

Table 2: Quantifying the inefficiency of discretization: Each of the three dimensions, $(x, y, \psi)$ is discretized into a grid of size $g$, so that the domain is partitioned into $g^3$ cubes. We present the fraction of the cubes for which all eight corners of this cube do not evaluate to the same prediction, which is a sufficient condition for the cell to intersect with a decision boundary.

This idea is formalized by Julian & Kochenderfer (2019) as Algorithm 1. Because multiple advisories will be issued whenever a cell straddles the decision boundary, the discretized algorithm will wrongly include some states as reachable since a worst-case analysis needs to take account of *all* reachable states. Table 2 gives an indication of the magnitudes of overestimation, presenting how much of the state space will lead to multiple advisories under a simple discretization scheme.

Julian & Kochenderfer (2019) do not use an equispaced grid, but the basic point – that discretization error cannot be made negligible – is an inescapable feature of this approach. And any false positives in a single-step decision function will be amplified in the dynamic analysis, as more reachable states at one point time lead to even more reachable points at the next step, so a 1% overestimation at one step may be compounded to considerably more through the dynamics. Coincidentlly, Julian & Kochenderfer (2019) are able to reach a usable solution, but are unable to guarantee the absence of near collisions under some realistic parameter configuations.

Note how the cells can be traversed in any order. This is a simple way to see that this algorithm is not fully using the spatial structure of the problem. Next, we incorporate this knowledge.

---

**Data:** Maximum distance set $\mathcal{R}_0$, policy $f$, an "unsafe set" $U$, transition dynamics $\Psi$,
       encounter length $T$.
**Result:** Guaranteed to not reach an unsafe state from $\mathcal{R}_0$ under policy $f$?
initialization: $t = 0$, done = False;
Partition the state space into cells $c \in \mathcal{C}$;
**while** *not done* **do**
    $t = t + 1$;
    $\mathcal{R}_t = \emptyset$;
    **for** $c \in \mathcal{C}$ *such that* $c \cap \mathcal{R}_{t-1} \neq \emptyset$ **do**
        **for** $i$ *such that* $f(c) \cap \{x : x_i \geq x_j \text{ for } j \neq i\} \neq \emptyset$ **do**
            **for** $c' \in \mathcal{C}$ *such that* $c' \cap \Psi(i, c) \neq \emptyset$ **do**
                $\mathcal{R}_t \leftarrow \mathcal{R}_t \cup c'$
            **end**
        **end**
    **end**
    done = $\mathcal{R}_t == \mathcal{R}_{t-1}$ or $U \cap \mathcal{R}_t \neq \emptyset$ or $t > T$.
**end**
Return $\mathcal{R}_t \cap U == \emptyset$

**Algorithm 1:** Algorithm from Julian & Kochenderfer (2019) for computing whether an unsafe set $U$ can be reached under a policy $f$ beginning from $\mathcal{R}_0$ under transition dynamics $\Psi$.

### 3.3.3 OUR PREIMAGE-BASED ALTERNATIVE

Rather than looping first the domain, then over actions at those points, Algorithm 2 loops over actions and, using the preimage, computes all reachable points under that action.

---

**Data:** $\mathcal{R}_0$, $f$, $U$, $\Psi$, $T$.
**Result:** Guaranteed to not reach an unsafe state from $\mathcal{R}_0$ under policy $f$?
initialization: $t = 0$, done = False;
**for** $i = 1, 2, \ldots, n_L$ **do**
    $\Xi_i = f^{-1}(\{x : x_i \geq x_j \text{ for } j \neq i\})$
**end**
**while** *not done* **do**
    $t = t + 1$;
    $\mathcal{R}_t = \emptyset$;
    **for** $i = 1, 2, \ldots, n_L$ **do**
        $\mathcal{R}_t \leftarrow \mathcal{R}_t \cup \Psi(i, \Xi_i \cap \mathcal{R}_{t-1})$;
    **end**
    done = $\mathcal{R}_t == \mathcal{R}_{t-1}$ or $U \cap \mathcal{R}_t \neq \emptyset$ or $t > T$.
**end**
Return $U \cap \mathcal{R}_t == \emptyset$.

**Algorithm 2:** Our preimage-based, exact algorithm for computing the dynamically reachable states in an ACAS.

While Algorithm 2 is exact – it will never wrongly say that a state can be reached – the accuracy of Algorithm 1 is ultimately controlled by the number of cells, $|\mathcal{C}|$. This is because it is necessary to perform $n_L$ verifications for each reachable cell, and the number of reachable cells is proportional to $|\mathcal{C}|$. Let $V$ denote the cost of a verification. Verification is known to be NP-complete (Katz et al. (2017)), so $V$ dominates all others calculation such as computing intersections or evaluating $\Psi(i, c)$. Thus, the computational cost of Algorithm 1 is $O(|\mathcal{C}|V n_L)$. In Algorithm 2 must initially compute $n_L$ preimages which dominates the entire calculation, which consists of relatively fast operations – applying the dynamics and computing intersections up to $T$ times, for $T$ a number around 40.

Let $P$ denote the cost of computing a preimage, then Algorithm 2 is $O(P n_L)$. So whilst it dispenses with the need to solve $O(|\mathcal{C}|)$ verifications, but may be more intractable if $P$ is significantly higher than $V$. Let the dimensions of the nonlinear layers in a DNN be $n_{\ell_i}$, then because in the worst case it is necessary to check each nonlinearity, each of which can be independently in a negative or positive configuration, $V = O(2^{\sum_i n_{\ell_i}})$. *Exact verification for even a single cell is impossible at present for*

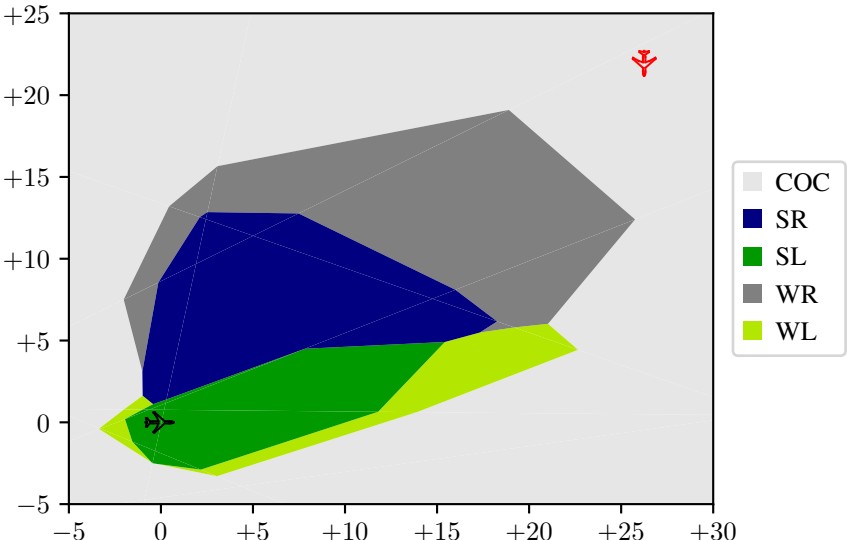

Figure 4: An encounter plot showing the optimal action at each $(x, y)$ distance configuration for a fixed angle of approach, indicated by the perpendicular orientation of the red (intruder) aircraft. Distances are measured in kilofeet.

*large networks*. We believe that preimages can be computed roughly (within a small constant factor) as easily as a verification – $P = O(V)$. We are currently developing this conjecture formally, the idea is that, as shown in Lemma 4, each nonlinear layer $\ell_i$ generates up to $2^{n_{\ell_i}}$ sets, the preimage of which must be computed through earlier layers.

In any case, as is true of any exponentially hard problem, the practical tractability of both $P$ and $V$ hinges importantly upon theoretical arguments showing that not all $2^n$ configurations of the nonlinearities of an $n$-dimensional layer can be achieved (Serra et al. (2017); Hanin & Rolnick (2019), and clever implementations that take account of the structure of the problems (e.g. Tjeng et al. (2017); Katz et al. (2017)).

The distinction between the two algorithms is made clearer by examining an *encounter plot* such as Figure 4. Encounter plots are concise summarizations of the policy function, here depicting the possible advisories, for a fixed angle of approach (which is here conveyed by the orientation of the red aircraft relative to the black). This figure, which replicates Figure 4 of Julian & Kochenderfer (2019), differs from it in a crucial respect: it is depicts the *analytically-computed* preimage of the five sets where each of the advisories are issued (details of the experiment are in Section D.3). The shaded areas arise from plotting polytopes, as in Algorithm 2. Julian & Kochenderfer (2019), on the other hand, produce such plots by evaluating the predictions of the network on a fine grid. The different manner in which the plots are produced is an exact analogue of the different way that the networks are summarized and analyzed through time.

## 4 CONCLUSION

In many areas, safety and interpretation inhibit the use of DNNs, because their use still requires a good deal of indirect experimentation and oversight to have confidence that it will not act in an unintuitive way. This paper has proposed computing the preimage of the decisions of a DNN as an intuitive diagnostic that can help to anticipate problems and help domain experts gain trust in a DNN, even if they are unable to formally articulate what makes a DNN trustworthy. In order to do this, we developed the preimage of a DNN and presented an algorithm to compute it. We demonstrated the utility of the preimage to understand counterintuitive behavior from a cart pole agent, and to more precisely characterize the set of states that would be reachable in an existing application of DNNs to aircraft automated collision avoidance systems.

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

# A    PROOFS

## A.1    PROOF OF LEMMA 1

*Proof.*  Unroll Equation 1. Let $S \subseteq \mathbb{R}^{n_{\ell+k}}$ be arbitrary.

$$
\begin{aligned}
&(f_{\ell+k} \circ f_{\ell+k-1} \circ \ldots \circ f_\ell)^{-1}(S) \\
=&\{x : (f_{\ell+k} \circ f_{\ell+k-1} \circ \ldots \circ f_\ell)(x) \in S\} \\
=&\{x : f_{\ell+k}((f_{\ell+k-1} \circ f_{\ell+k-2} \circ \ldots \circ f_\ell)(x)) \in S\} \\
=&\{x : (f_{\ell+k-1} \circ f_{\ell+k-2} \circ \ldots \circ f_\ell)(x) \in f_{\ell+k}^{-1}(S)\} \\
=&\vdots \\
=&\{x : (f_{\ell+1} \circ f_\ell)(x) \in (f_{\ell+2}^{-1} \circ \ldots \circ f_{\ell+k-1}^{-1} \circ f_{\ell+k}^{-1})(S)\} \\
=&\{x : f_\ell(x) \in (f_{\ell+1}^{-1} \circ f_{\ell+2}^{-1} \circ \ldots \circ f_{\ell+k-1}^{-1} \circ f_{\ell+k}^{-1})(S)\} \\
=&(f_\ell^{-1} \circ \ldots \circ f_{\ell+k-1}^{-1} \circ f_\ell + k)^{-1}(S).
\end{aligned}
\tag{5}
$$

$\square$

## A.2    PROOF OF LEMMA 2

*Proof.*

$$
\begin{aligned}
&x \in f^{-1}(\cup_{i=1}^N S_i) \iff \\
&f(x) \in \cup_{i=1}^N S_i \iff \\
&f(x) \in S_1 \text{ or } f(x) \in S_2 \text{ or } \ldots \text{ or } f(x) \in S_N \iff \\
&x \in f^{-1}(S_1) \text{ or } x \in f^{-1}(S_2) \in S_2 \text{ or } \ldots \text{ or } x \in f^{-1}(S_N) \iff \\
&x \in \cup_{i=1}^N f^{-1}(S_i).
\end{aligned}
$$

$\square$

Note that an identical argument shows that $f^{-1}\left(\cap_{i=1}^N S_i\right) = \cap_{i=1}^N f^{-1}(S_i)$. This can be useful in some applications where where $S_i$ can be wrtten as $\Psi \cap \Xi_i$ – writing $\cup_i S_i$ as $\Psi \cap \cup_i \Xi_i$ may be more efficient.

# B    THE INVERSE OF A RESIDUAL BLOCK

The key function in a residual block is

$$
x \mapsto W_2 \text{ReLU}(W_1 x) + x.
$$

Combining arguments similar to Lemma 3 and Lemma 4, we have that

**Lemma 5** (Preimage of residual block)**.**

$$
\begin{aligned}
&(z \mapsto W_2 \text{ReLU}(W_1 z) + z)^{-1}(\{x : b - Ax \geq 0\}) \\
=&\{x : b - A(W_2 \text{ReLU}(W_1 x) + x) \geq 0\} \\
=&\bigcup_{\nu \in \{0,1\}^n} \{x : b - A(W_2 \text{diag}(\nu)W_1 + I)x \geq 0, -\text{diag}(1 - \nu)W_1 x \geq 0, \text{diag}(\nu)W_1 x \geq 0\}.
\end{aligned}
\tag{6}
$$

# C    COLLECTING THIS ALL UP

Section 2.1, Section 2.2, Section 2.3, and Section 2.4 together give us a recipe for inverting a wide class of image sets (region-unions) for a wide class of DNNs (those which can be written as the composition of linear and ReLU functions). To summarize the steps are:

1. Put the network into "standard form":
   (a) Embed any transformations that are "off" at inference time, such as dropout or batch normalization into the weights.
   (b) Rewrite the network in flattened form, for example replacing $3 \times 32 \times 32$ tensors by $3072 \times 1$ vectors. This is a convention to facilitate our polytope formulation.
   (c) Rewrite all transformations as compositions of linear and ReLU functions. For example, convolution and average pooling are linear functions. Maxpooling, hard tanh, and leaky ReLU can be written as the composition of linear and ReLU functions.

2. Let $f = f_L \circ f_{L-1} \circ \ldots \circ f_1$ denote the network in this form.

3. Let $R_L = \cup_i \Delta_i$ be the image set that we wish to invert, for example $R_L = \Delta_1 = \{x : x_1 \geq x_2\} \subseteq \mathbb{R}^2$ in a binary classifier.

4. Compute $f_L^{-1}(\Delta_i)$ for all $i$, using Lemma 3 or Lemma 4.

5. Each term above is a region-union, thus $\cup_i f_L^{-1}(\Delta_i)$ is a region-union.

6. By Lemma 2, $R_{L-1} \triangleq f_L^{-1}(R_L) = \cup_i f_L^{-1}(\Delta_i)$.

7. $R_{L-1}$ is a region-union, so apply the same argument to compute $R_{L-2} \triangleq f_{L-1}^{-1}(R_{L-1}) = f_{L-1}^{-1}(f_L^{-1}(R_L))$.

8. Repeat for $\ell = L - 2, \ldots, 1$ to compute $R_0 = f_1^{-1}(R_1) = \ldots = (f_1^{-1} \circ f_2^{-1} \circ \ldots \circ f_L^{-1})(R_L)$.

9. Appeal to Equation 1 to conclude that $R_0 = f^{-1}(R_L)$.

## D  DETAILS OF EXPERIMENTS

### D.1  SECTION 3.1

The dataset of 500 observations is generated using the scikit-learn function `sklearn.datasets.make_moons` with `noise = .2`.

Weights are initialized according to a uniform$\left(-\sqrt{\text{in features}}, +\sqrt{\text{in features}}\right)$ distribution (the PyTorch default), and were run for 1000 epochs of with a batch size of 128. Gradient steps were chosen by the adam optimizer (Kingma & Ba (2014)) with learning rate of 0.005 and $(\beta_1, \beta_2) = (0.9, 0.999)$.

### D.2  SECTION 3.2

Our experiment is based upon the "CartPole-v1" environment from Brockman et al. (2016). The fitting procedure is based upon the Monte Carlo policy gradient vignette from the PyTorch project,[1] with a considerably simplified control policy, consisting of DNN wih only five hidden units.

#### D.2.1  VELOCITY MAGNITUDE

As far as we can tell, there is no single best methodology for computing limits on the velocities $(\dot{x}, \dot{\theta})$ in the cart pole problem. In premise, very high velocities could be supported by the discretization scheme, but these are unlikely to be achieved by any feasible sequence of actions. If we restrict attention to limits described by actions, how should we characterize the set of actions we consider? For example, should we simply observe the behavior of some non-optimized agent? Should we deliberate construct agents to pursue velocity-maxmizing strategies? Should we force agents to have the same initialization as that prescribed in the fitting?

We concluded that an interpretable baseline which gave quantitative bounds robust to details of parameterizations would be best. For this, we chose our limits on $\dot{x}, \dot{\theta}$ as the values that answer the question "how fast can the cart and pole be moving if we start from rest with the cart all the

---

[1] `https://github.com/pytorch/examples/blob/master/reinforcement_learning/reinforce.py`

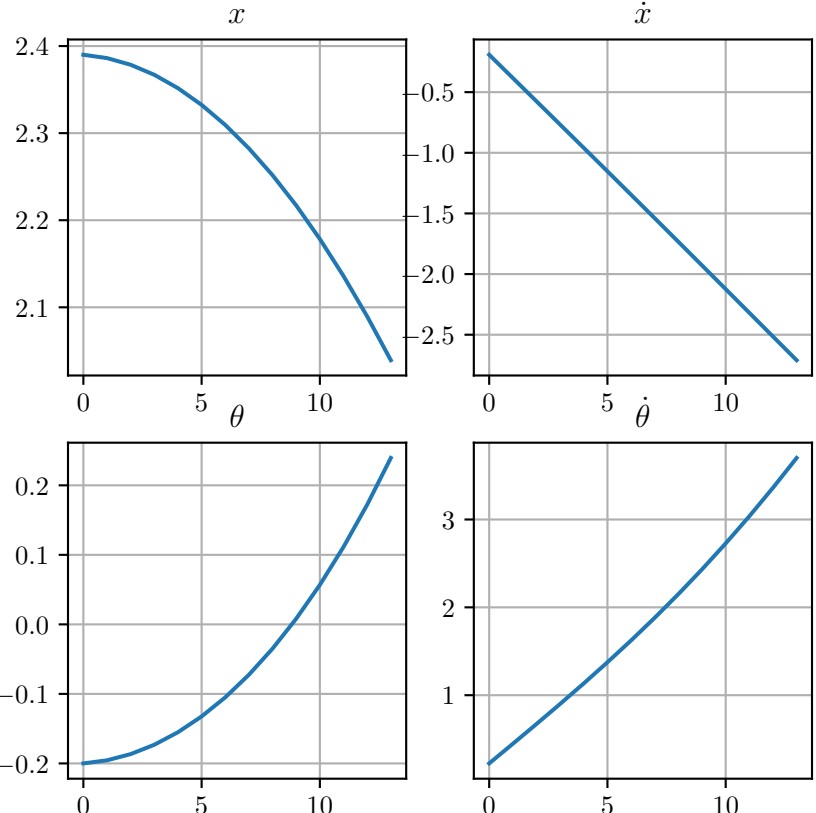

Figure 5: The evolution of the cart pole state under a policy which continually accelerates left until failure (when $\theta \geq \pi/15$), starting from rest, with the cart near the right boundary and the pole near its right boundary ($(x, \dot{x}, \theta, \dot{\theta}) = (2.39, 0, -.20, 0)$).

way to the right, and the pole all the way to the left, and continually push left until failure?". This experiment is plotted in Figure 5, where we see that the implied limits are $\pm 3.0$ and $\pm 3.5$ for $\dot{x}$ and $\dot{\theta}$, respectively.

In addition to being easy to envision and interpret, a policy that pushes in a uniform direction is a natural boundary between benign "dumb" policies and those that more actively seek to exercise worst-case scenarios through some deliberately degenerate behavior.

These limits largely agree with three the other candidates we considered:

- The same experiment, although constrained to obey the same initialization as in Brockman et al. (2016) (these limits are tighter, understandably, roughly 2.25, 2.75 respectively).

- A simple one-parameter agent which (starting from an initialization near the origin) seeks out high velocities by beginning to push in a uniform direction, then switches to the opposite direction.

- The most extreme values emitted from a small (and hence exhibitting more erratic behavior early on the in the fitting) DNN that is able to eventually achieve good performance.

## D.3 SECTION 3.3

The analysis presented in Section 3.3 is based entirely on data generated by Julian & Kochenderfer (2019)'s system that formulates and solves dynamic programs to deliver lookup tables of optimal collision avoidance behavior in the same manner as the FAA's proprietary software. Our DNN modelling is somewhat different, however, and whilst we think that our results can be interpreted within their framework, in this section we detail the aspects of our analysis that differ from Julian & Kochenderfer (2019)'s.

### D.3.1 FITTING – OPTIMIZATION CRITERION

The first manner in which our approach is different is the fitting criterion: Julian & Kochenderfer (2019) issue advisories as a function of position and velocities indirectly: by first fitting the continuation value to taking each action, and then choosing the action with the highest predicted continuation value. This oblique approach is understandable: this work is the continuation of an extended project to build (Kochenderfer & Chryssanthacopoulos (2011)) and compress the Q-table (Julian et al. (2016), Katz et al. (2017)).

And although the Q-values themselves have some interpretation, issuing advisories requires only knowing the greatest. We hypothesize that it is easier to solve a problem which recognizes an invariance of the prediction to any increasing transformation. And that is what we find – by replacing Julian et al. (2016)'s mean-squared-error-based criterion with cross entropy loss to directly model the optimal decision, we are able to achieve better performance with smaller networks. One statement of the improvement is that Julian & Kochenderfer (2019) use a five layer fully connected layers with 25 neurons each to achieve an accuracy of 97% to 98%. We are able to achieve comparable accuracy with a *two* layer 25 neuron fully connected network (a network of the same size targetting MSE loss only attains an accuracy around 93%).

Why is anything less than complete fidelity to the Q-table acceptable in an approximation? The answer seems to be twofold: firstly the Q-table is itself not perfect, because of discretization artifacts. One can observe physically implausible sawtooth-like decision boundaries that arise from a coarse grid in the top plot of Julian & Kochenderfer (2019) Figure 4. The second is that accuracy alone does not capture the genuine metric of goodness, for example in the bottom plot of Figure 4 of Julian & Kochenderfer (2019) we see a highly accurate network that exhibits unusual "islands" of SR completely encompassed by a region of WR that are both not present in the ground truth, and also prescibe a conceptually wrong relationship (a pilot could be initially advised a strong right turn, then after some period of lessened danger have it downgraded to a weak right, only to have it re-upgraded to a strong right, seemingly although the danger continues to lessen). The correct metric seems to rather be plausibility of the prescribed encounter plot. These observation leads us to not worry too much about small differences in model accuracies in favour of plausibility of the encounter plots.

### D.3.2 FITTING – SYMMETRY

The second manner in which our approach differs from Julian & Kochenderfer (2019) is in the domain being fitted. Julian & Kochenderfer (2019) fixed a lookup table over $(x, y, \psi) \in [-56000, +56000]^2 \times [-\pi, +\pi)$. However, if we let $Q : \mathbb{R}^3 \to \mathbb{R}^5$ denote the Q-function as a function of the state $s = (x, y, \psi)$, then the physics of the problem insure that

$$Q(T_i s) = T_o Q(s) \text{ where } T_i = \begin{pmatrix} +1 & 0 & 0 \\ 0 & -1 & 0 \\ 0 & 0 & -1 \end{pmatrix} \text{ and } T_o = \begin{pmatrix} +1 & 0 & 0 & 0 & 0 \\ 0 & 0 & +1 & 0 & 0 \\ 0 & +1 & 0 & 0 & 0 \\ 0 & 0 & 0 & 0 & +1 \\ 0 & 0 & 0 & +1 & 0 \end{pmatrix}.$$

This relationship clearly only works for $a_{\text{prev}} = \text{COC}$, but similar symmetries will exist more generally. Thus, strictly speaking, half of the lookup table is unneeded, and moreover it would seem wasteful to ask a network to learn (what we already know to be) the same thing twice. Thus, our method is to only fit $f$ over $(x, y, \psi) \in [-56000, +56000]^2 \times [0, +\pi)$, and when needed to infer $f(s) = T_o f(T_i s)$ for $s = (x, y, \psi)$ with $\psi < 0$. In so doing, we halve the data set size, but leave other data fitting parameter unchanged.

To continue the analysis above describing comparable performance from smaller networks, exploiting symmetry enables us to achieve accuracy above 97% from a one layer, 24 neuron network. For computational ease, Figure 4 is computed on a 16 neuron network that achieves about 96% accuracy.

### D.3.3 INVERSION – PROJECTION

Figure 4 was formed by taking the fitted $n_0 = 3$ DNN, fixing $\psi$ to a given value, and inverting the resultant $n_0 = 2$ DNN (if $W_1, b_1$ denote the weights and bias of the original DNN, then projecting onto the $W_1' = W_1 \begin{pmatrix} 1 & 0 \\ 0 & 1 \\ 0 & 0 \end{pmatrix}, b_1' = b_1 + \psi W_1 \begin{pmatrix} 0 \\ 0 \\ 1 \end{pmatrix}$.

## E ADDITIONAL COMPUTATIONAL DETAILS

Definition 1, as well as Lemma 3 and Lemma 4 worked with what is known as the *H representation* of a polytope. What Fukuda (2014) terms the "Minkowski-Weyl Theorem" states that there is an equivalent representation, as the Minkowski sum of (1) a convex combination of vertices, and (2) a conic combination of some rays.

**Definition 3** (Polytope (V representation)). *A polytope in $\mathbb{R}^n$ is a set that can be written as*

$$\left\{ \sum_i \lambda_i v_i + \sum_j \nu_j r_j : \lambda_i \geq 0, \nu_j \geq 0, \sum_i \lambda_i = 1 \right\}$$

*for some $v_i \in \mathbb{R}^n, r_j \in \mathbb{R}^n$.*

The Minkowski-Weyl Theorem assures us that Definition 3 is equivalent to Definition 1.

To introspect on DNNs it is handy to have both the H and V representation of the preimage. Unfortunately, computing the V representation from the H representation is computationally challenging, both theoretically and practically. Analyzing formally the computational complexity of the problem is technical, and also complicated by a host of special cases, but roughly speaking:

- Evidently the complexity of the problem depends not only on the size of the input, but also the output.

- Small inputs can have large outputs. For example, a cube in $d$ dimensions has an H representation with $2d$ rows, but $2^d$ vertices.

- In general, there are no known algorithms that have polynomial time complexity in the input and output size.

- For some polytopes, there are algorithms that have polynomial time and space complexity in the input and output. However, but this is still exponentially large in the dimension of the polytopes, again because output size can be exponentially large in the input size.

The more precise statements of this summary can be found in Fukuda (2014), Section 9.

Our problem may lie in some more easily-solved subclass of problems (though the fact that even simple geometric objects like cubes exhibit exponential growth of the output as a function of the input dimension makes this perhaps less likely). Our review of the theoretical literature was unsuccessful int this regard, and empirically we observed empirically that the runtime of our calculation did rise at an exponential-like rate with the dimension of the input. Thus: we did not find the naïve approach of computing the H representation of the preimage, then computing from that the V representation to be practical.

Happily, when $W$ is full rank, we can give the V representation corollary of Lemma 3.

**Lemma 6.** *Suppose that $W$ is full-rank, and let $W^\dagger$ be its pseudoinverse and $W^\perp$ be a basis for the nullspace (with $k$th column $W_k^\perp$) then, for $\lambda_i \geq 0, \nu_j \geq 0, \sum_i \lambda_i = 1$:*

$$
\begin{aligned}
Wx + a = \sum_i v_i \lambda_i + \sum_j r_j \nu_j &\iff \\
x = \sum_i (W^\dagger v_i - a)\lambda_i &+ \sum_j (W^\dagger r_j)\nu_j + \sum_k W_k^\perp \gamma_k \\
= \sum_i (W^\dagger v_i - a)\lambda_i &+ \sum_j (W^\dagger r_j)\nu_j + \sum_k W_k^\perp \gamma_k^+ + \sum_k (-1 \times W_k^\perp)\gamma_k^- \\
\text{where } \gamma_k^- \geq 0, \gamma_k^+ \geq 0.
\end{aligned}
\tag{7}
$$

*This is a V representation with vertices $(W^\dagger v_i - a)$ and rays $(W^\dagger r_j), W_k^\perp, -1 \times W_k^\perp$.*

Checking the rank of $W$, computing the pseudoinverse, and computing a basis for the nullspace are all quick and standard linear-algebraic routines.

For example, in our experiments were were unable to come anywhere near computing the V representation of a an MNIST classifier ($n_0 = 784$) directly using a standard software such as *cdd* (Fukuda & Prodon (1996)). However, for networks with $n_{\ell-1} \geq n_\ell$ for all $\ell \geq 1$ (which in our simple fitting procedure, was sufficient to insure that $W$ would be full-rank), it was quite possible using Lemma 6: (1) compute the V representation of the polytopes to be inverted (these will be ten dimensional, and highly structured), then (2) apply Equation 7 iteratively backwards.

### E.1    POLYTOPE DIMENSION

One slightly subtle point is that the terms in Equation 4 overlap at the boundaries, for example if $b \geq 0$ then the origin is in fact contained in every term in the union. In this work, we only form *full dimensional* polytopes, roughly those which have strictly positive $n$-dimensional volume. Removing lower-dimensional sets does not change the union, so this does not substantively change the preimage. A study which was more explicitly interested in the details of decision boundaries might not want do make such a restriction. Some more discussion of this point is in Section E.1.

We mentioned in Section 2.4 that our analyses restrict attention to polytopes that have full dimension. Formally, the dimension of a polytope is the maximum number of affinely independent points in that polytope, minus one. And a full dimensional polytop in $\mathbb{R}^d$ is one with dimension $d$. Geometrically, sets which are not full dimensional lie on the boundary between sets, and if they have a nonempty preimage, it will lie on a boundary shared by another polytope, which does have full dimension.

This idea is made clearer by explaining how we check if a polytope has full-dimension. Algorithm 8.4 from Section 8.3 of Fukuda (2014) states that $\{x : b - Ax \geq 0\}$ is full-dimensional iff the optimization problem

$$
\text{maximize } \varkappa \text{ subject to } Ax + \mathbf{1}\varkappa \leq b, \varkappa \geq 1.
\tag{8}
$$

achieves a positive criterion. Here $\mathbf{1}$ is a conformable column vector of ones, representing the intuition that it is possible to loosen all inequality conditions by a strictly positive amount, meaning that there is some volume interior to the polytope. The ancillary condition that $\varkappa \leq 1$ is used to keep the problem well-conditioned.

Empirically, *most* sets in the region-union comprising Equation 4 are not full dimensional, and as soon as we know that an element of a preimage region is not full dimensional, we need not consider it anymore. Future will will use more careful analysis to detect analytically when sets must necessarily be less than full dimensional, but for now we query each of the $2^n$ subsets of the region-union. Thus most of our computational time is spent in calculations of the form of Equation 8. We tested both Mosek and Gurobi as software for solving linear programs such as Equation 8, and found Gurobi to be faster.

| Hidden layer widths | Accuracy | # parameters | Storage (MB) | Time (s) |
|---|---|---|---|---|
| 8 | 0.973 | 42 | 1.105 | 0.205 |
| 12 | 0.975 | 62 | 35.256 | 2.907 |
| 16 | 0.974 | 82 | 985.767 | 52.912 |
| $4 \rightarrow 4$ | 0.972 | 42 | 0.260 | 0.269 |
| $6 \rightarrow 6$ | 0.967 | 74 | 6.764 | 3.205 |
| $8 \rightarrow 8$ | 0.971 | 114 | 42.089 | 26.256 |
| $4 \rightarrow 4 \rightarrow 4$ | 0.969 | 62 | 3.125 | 1.802 |
| $6 \rightarrow 6 \rightarrow 6$ | 0.968 | 116 | 279.138 | 117.248 |
| $4 \rightarrow 4 \rightarrow 4 \rightarrow 4$ | 0.973 | 82 | 43.695 | 27.253 |

## E.2 POLYTOPE VOLUME

In low dimensions, computing the volume of a polytope from its V form is fast and space efficient. In the analysis presented in Section 3.2, we used the Qhull software which implements the Quickhull algorithm (Barber et al. (1996)) via `scipy.spatial.ConvexHull`. This computation scales poorly with dimension, however, for instance in our experiments it stopped being usable around ten dimensions. Büeler et al. (2000) gives some "why" and "how" about this difficulty.

In high dimensions analysis of volumes seems difficult. However, if we already plan to compute the V representation of the preimage – also a highly complex operation – we may be able to compute the volume "for free". Avis (2000) shows how to compute the volume as a byproduct of his algorithm for converting between H and V representation (solve the vertex enumeration problem), and it seems quite plausible that the same ideas could be adapted to other algorithms that solve the vertex enumeration problem as well. We hope to investigate this further and possibly incorporate it into our software.

## E.3 CLOCK TIME TO SOLVE PROBLEMS OF VARYING SIZES

In this section, we give a sense of the computational difficulty of inverting a DNN. The general finding is that for simple problems, one layer networks of around 16 neurons, two layer networks of about 8 neurons apiece, or three layer networks of about six neurons apiece are easily inverted on a low-powered laptop, but the rate of growth is empirically very fast. Certainly even the five layer, 25 neurons apiece networks employed in Julian & Kochenderfer (2019) are out of reach under the current implementation.

We do not use any multiprocessing, though the essential computation, checking polytope emptiness, is embarassingly parallel.

We perform ten fittings and compute preimages per size, and report the average model accuracy, time to compute the preimage, and standard deviation of each. To get a sense of the space complexity, we also present the disk storage necessary to hold both the H and V forms of a complete preimage partitiion (pickled dense numpy arrays of `float64s`). Since we do not re-tune hyperparameters across runs, accuracy is solely indicative.

All timings were performed on a 1.6 GHz Dual-Core Intel Core i5 CPU.

A more formal analysis of the complexity of the computation will follow in future work, as will speed improvements from a more sophisiticated logic for handling empty preimage regions, along with further experimentation on "tricks" such as sophisticated regularization or clever initialization schemes that might enable greater modelling capacity without increasing the scale of the network (e.g. Zhou et al. (2016) or Xiao et al. (2018)).

The problem is as described in Section 3.1, with further detail given in Section D.1.

