# OpenReview forum: "Computing Preimages of Deep Neural Networks with Applications to Safety"
_ICLR.cc/2021/Conference — Reject_

### Official Review · AnonReviewer1 · 2020-10-26
**Not convinced that pre-image computation offers any realistic advantage**

**Rating:** 2
**Confidence:** 5

**Review:**

Deep neural networks are known to be brittle, and can lead to dangerous consequences if left unverified. Forward reach set computation can be used as a basic primitive to verify properties of deep neural networks used in a robotic setting. There has been a rising interest in verifying larger neural networks used in safety critical setting.
 In this paper,  the authors propose a way to compute reachable sets for a neural network in a backward sense. Starting from the outputs of the neural network, and  then work it's way to the inputs. This is an interesting way to look at the problem itself,  but as the authors point out it is an intractable problem.

My concern about this paper is I don't see the use of a pre-image computation algorithm as being very useful. A forward reachability tool works pretty well for the size of neural networks considered in the paper. Pre-image computation does not provide any advantage in terms of scalability, as is apparent from the experiments. Moreover, almost any safety constraint that needs to be verified with system dynamics in the loop always should ideally work forward in time. Thus for the neural network controller from the inputs to the outputs.

Cartpole example : The authors come up with rules about, which output behaviors are correct for a few of the input regions. Then use this as a specification for the verification algorithm. But the very specifications, comes from reasoning about the forward behavior of the system dynamics itself. The idea of forward reach sets computation would generalize much better to a wide range of examples therefore.  Without the need to come up with such handcrafted rules.

The authors do make a convincing case for the ACASXu example. But this example is less interesting given the amount of attention it has received recently.

---

> ### Author Response · Authors · 2020-11-17
> **Reachability is quite related (see Table 1 of the paper), but cannot do somethings that preimages can.**
>
> We're not experts in reachability, so apologies if we mischaracterize it below.
>
> We agree that our method is not more scalable than a single reachability calculation. The relative computational efficiency of forward and backwards approaches for multi-step dynamics seems complicated to analyze in general, but here's our take on it. With forward reachability analysis, it appears necessary to solve a separate, costly verification at each step (though possibly there are more efficient ``updating'' methods for the forward calculation that we are not aware of). By contrast, with the backwards approach, _it is necessary to only compute the preimage once_ (see Algorithm 2). Then, given the preimage, stepping through time amounts to intersecting the reachable set with the preimage to compute actions, then applying the dynamics.
>
> What is something that is much easier (perhaps _only_ possible) by reasoning backwards? NN-based ACAS can sometimes exhibit ``islands'' of one prediction surrounded by regions of a different prediction (see Figure 4 of https://arxiv.org/abs/1912.07084) that are implausible on optimality principles. Reasoning backwards, one can simply look for disconnected segments. How would reasoning forward identify such behavior short of exhaustively partitioning the domain?
>
> There seems to be some confusion about the goal of the cartpole example. We are not attempting to prove or disprove some property of the network. Rather, we point out a particular instance of counterintuitive behavior, and then use the preimage to interrogate the extent of its occurrence. We are not particularly in transition dynamics, except as an informal explanation of why the counterintuitive behavior is evidently wrong. Our analysis in this regard is entirely static, and we do not analyze the set of states that are theoretically reachable over time.
>
> It's true that our formulation of wrong behavior is ````````''handcrafted''. This is by design, our motivation is to enable users to compare their expectations with the observed behavior of networks. Reasoning from inputs to outputs can help to answer well-formed questions, and this is important in safely interoperating with NNs. Reasoning backwards, however, can help to anticipate problems that are not well-formed. Many properties are obvious once observed, but not easily envisioned beforehand. Custom tools, such as projections of NN preimages onto two or three dimensions can help humans to identify behavior that they recognize as wrong when presented with it, but may not able to foresee.

---

### Official Review · AnonReviewer4 · 2020-10-26
**Paper needs substantial improvements.**

**Rating:** 3
**Confidence:** 4

**Review:**

The paper presents a method to verify if a NN is performing as expected in sensitive applications. Although the general area is very important in machine learning, the paper is not very well presented: The problem is not well stated, the approach is not very clear, and the results are not well justified.

- The presentation and the writing of the paper should be improved. Unfortunately, with the current format it is hard to glean the idea of the paper. There are some typos (e.g., 'bu' in page 3, 'plot' in page 4, etc.).
- There are some concepts that are not defined early on and maybe never in the paper. For example, what is the problem that the paper tries to solve mathematically? It is not very clear. What is the mathematical definition of a preimage?
- The authors say: "Preimages are most insightful and useful when the inputs and outputs have definite interpretation –
application areas where the need for massive networks is less". Its hard to fully understand but it seems that the method suffers scalability issues. Can this be formally analyzed? What is the complexity of the algorithm in time and space? Why is there a scalability issue? Is it a fundamental problem? How does this limit the scope and applicability of the method? Also, what does "definite interpretation" mean?
- The NN used in the experiments are very tiny. I would consider experiments that reflect more realistic situations in the real-world. Current setup significantly limits the scope of the method.
- It is not clear how to verify the performance of the method. Results in Figure 1 and 2 does not show us the quality of the method, is it doing good or bad? I found the results in Figure 1 surprising as the moon data is fairly symmetric while the preimage is biased towards one class. Is there a reason for that?

---

> ### Author Response · Authors · 2020-11-17
> **Idea: Compute the set of all inputs that a DNN will assign some class label and argue that this can be an important diagnostic.**
>
> Thanks for your concrete concerns. I'll skip the first bullet point since it is subsumed by the others. To address the rest:
>
>   * The problem we want to solve is addressed by Table 1. Mathematically, given a DNN f and a subset of the range, Y. We want to compute the set of all inputs that evaluate to a value in Y under f, denoted f^{-1}(Y). For example, if f is an MNIST classifier, and Y is the 10-dimensional subset of Euclidean space where the first coordinate is highest, then f^{-1}(Y) is the subset of the 784-dimensional unit cube of all grayscale images that will be classified as a "0". This table contrasts our work with a fairly large related literature in verification and inversion.
>   * "Definite interpretation" means problems where the inputs have some meaning independently of the model and are expected to satisfy some known properties. For example, models in engineering or robotics are constrained by interpretable, fundamental relationships from physics. This is different to work in domains like computer vision, sound, or natural language, where human intuition cannot be simply applied to introspect on a model's working. Networks applied to problems with definite interpretation (such as those in Julian and Kochenderfer 2019) tend to be smaller.
>     We discuss the complexity of the algorithm after Algorithm 2 (Section 7). Put simply: it is NP-complete, as are exactly solutions of many of the related problems discussed in Table 1. It is a fundamental problem for general networks, but there is considerable work "tricks" to make the network easier to analyze with minimal loss in performance (such as "Training for Faster Adversarial Robustness Verification via Inducing ReLU Stability" Xiao et al 2019).
>   * This is largely addressed above, but to reiterate the use case that we envision is not a large network that achieves state of the art performance on a standard computer vision test problem from deep learning. It's wrong to equate scale with "realistic". Though the network is small, we've presented a "real world" application to aircraft collision avoidance systems (NASA have run real tests of real-world test pilots flying real planes to test the system that is being compressed by neural networks, see https://ntrs.nasa.gov/citations/20150008347).
>   * There's no general notion of quantifiable "performance" of the method -- inherently it tries to help _people_ understand how their networks behave. Given a problem instance, it can be used to develop metrics of how well a network is performing that can't be discussed otherwise, such as the volume of the input space that exhibits wrong behavior discussed in Section 3.2. On the two moons data, the asymmetry of the decision boundary is basically noise in regions of the domain where there is not much data, if you fit the network starting from another seed, you might see a quite different pattern in the top right and bottom left corners.

---

### Official Review · AnonReviewer2 · 2020-10-27
**The quality of the paper needs to be improved**

**Rating:** 4
**Confidence:** 3

**Review:**

There are many issues in the paper that can be improved.

The title is not appropriate, this work does not address safety applications. It is worth noting that the word safety is not defined and not used in the main body of the paper.

It is difficult to follow the presentation of the paper since mainly the applications are presented and then some contributions given, in the same presentation as the abstract.

A major issue it that the paper is missing some important theoretical analysis. Of particular interest is the existence of the preimages, because not all outputs have inputs. Moreover, the uniqueness of the solution needs to be studied. These properties should depend on the used nonlinearities and architecture of the neural network.

There are many spelling and grammatical errors, such as “suprising”, “have been been”, “Coincidentlly”, “configuations”

---

> ### Author Response · Authors · 2020-11-17
> **Clarifying theoretical analysis**
>
> Our point was not to take a strong stance on what "safety" entails, we are happy leaving this as an informal notion. We will clarify a bit in the main body of the paper. If a deeper justification is needed, we'd simply observe that our main application concerns a paper titled "Guaranteeing safety for neural network-based aircraft collision avoidance systems.".
>
> We are sympathetic about the presentation. Our work is not a nuanced insight about neural networks, nor is it a method that addresses an established problem. Rather, we present a method to let _users_ of DNNs understand their problem domain. For example, a main theme arising from a meeting of the United Nations Institute for Disarmament Research was the need for better metrics and tools to allow lawyers, ethicists, soldiers, and developers to reason about the use of lethal autonomous weapons systems (https://meetings.unoda.org/meeting/laws-webinars-2020/).
>
>
> To your questions about the theoretical analysis. I think that all of your concerns are addressed, but I appreciate that it may not be clear. I gather from the way that the question is phrased that it is not entirely clear that we are mainly proposing the preimage of class labels, meaning: computing the set of all inputs that receive some classification (for more on this, please see my response to AnonReviewer3, also Table 1). Hopefully that alone clarifies a lot, but here's some more detailed feedback.
>
> (1) Preimages may not exist -- it is perfectly valid to ask for the preimage of some subset of the range that has no inputs that lead to it -- but the presentation and code will return an empty preimage. But this is tautologically true and covered adequately by the existing notation. In any case, this is only true in some technical sense: the preimage of a class that has a correctly-labeled data point will, by construction, exist.
>
> (2) The preimages are _unique_. A simple way to see this is that the preimage of every class label is equivalent to the whole network, so far as classification is concerned. This is because for any input, we figure out which preimage it lies in, and assign it that label, in lieu of a forward pass. For a fixed instance of the network, thus, there cannot be two preimages that represent it.
>
> (3) There's no dependence of the above two points on the archiecture or activations, so long as they satisfy our assumptions: the forward pass of the pass is the composition of linear and relu functions. Our argument is twofold: within this class, everything we wrote is correct (we've asserted and proved everything that we think is needed?), and (2) this class actually includes many of the "layers" used in practice. For example, max-pooling can be written as the composition of linear and relu functions.
>
> Sorry if this was not conveyed effectively. Happy to have your feedback on how this can be improved!

---

### Official Review · AnonReviewer3 · 2020-10-30
**interesting discussion of using network pre-images for interpretation**

**Rating:** 3
**Confidence:** 4

**Review:**

The paper details a way of investigating the space of pre-images that lead to a particular output from a ReLU layer, with the goal of using the inverted representations as a way to understand the deep neural networks. Three experiments are proposed where the authors claim that the computed pre-images help interpret the network decision making.

Overall the paper is interesting, however I am not certain of the novelty as some related work is not discussed. Additionally, although the practical application of the method is interesting, the clarity could be improved for the last experiment.

Positives:
* Understanding the invariances of neural networks can potentially lead to more interpretable models, and one way to investigate this is by looking at the preimages for a network.
* The paper is a nice mix of theoretical results which lead to practical applications

Questions and Concerns:
* The authors state that maxpool can be rewritten in terms of a linear component and a ReLU, but this is non obvious. If this is true, a mathematical formulation should be explicitly included in the paper.
* The paper is missing some potentially related references. Previous work has investigated how multiple stimuli can get mapped onto (approximately) the same point in recognition networks by inverting the representations via iterative gradient descent (Mahendran & Vedaldi 2015, and recent work including invarianced based adversarial examples in Jacobsen et al. 2019 or model metamers in Feather et al. 2019). How does the proposed preimage computation help improve model interpretability beyond this previous work, especially given the authors statement that the method is intractable for large networks?
* The paper does not discuss invertible networks which have a bijective mapping between the input and output (ie Invertible Residual Networks in Behrmann et al. 2019). Discussing this work seems relevant if the goal is to make models such that one can start with hypothetical outputs and understand the inputs that lead to them.
* The final example of using this method in practice for ACAS systems is interesting, but it is difficult to follow what “success” would mean for this experiment

Minor points:
* The following sentence on page 3 seems to be missing something “Preimages are most insightful and useful when the inputs and outputs have definite interpretation – application areas where the need for massive networks is less.”.
* There it a typo in the last sentence of page 3 (“bu”->”but”)

---

> ### Author Response · Authors · 2020-11-17
> **We compute the preimage of entire subsets of the codomain, not just a single point.**
>
> Thanks for the pointer to Jacobsen et al 2019, it's a relevant paper that we weren't aware of. We agree with your general framing in terms of invariance, and we discuss briefly Behrmann et al 2018 and a related paper by Carlsson et al 2017. Compared to these papers, we say less about more: we examine only the "argmax pre-image" (in the terminology of Jacobsen et. al. 2019), and are able to only conduct more abstract analysis.
>
> We make this distinction in the last two lines of Table 1. The point being that domain-experts such as policymakers or scientists cannot productively work with the preimage of a single logit to answer more abstract questions from their fields. Our goal is not to use the preimage to show deep learning researchers some fundamental insight about neural networks, rather but to propose a tool that enable users to better interoperate with their instance of a neural network.
>
> Scaling up the idea of Behrmann et al and Carlsson et al from a single point in the range to full-dimensional subsets is not too difficult, but neither is it trivial. More to the point, no one had done it that we were aware of, and there are interesting questions that can only be addressed with it (for example, computing the volume of a particular variety of wrong behavior in the cart pole application). We use the exact preimage of the class labels to give an approach to verifying dynamic properties that sidesteps the need for discretization completely, this could not obviously be done with the preimage of a single logit? It's true that we only propose, but do not carry out, the full ACAS reachability exercise with our alternative, and that the paper would be stronger if we did.
>
> The connection to invertible resnets is an interesting one, and one that we are actively working on now. You will see in Appendix B that we present the preimage of a residual block, and we are using similar methods to develop a regularization method for fitting models for which it is easier to compute the preimage. Nonetheless, as far as we can see, while invertible resnets make it trivial to compute the pre-image of a single point, there is no reason that their structure would help controlling the combinatorial issues that arise when computing the pre-image of sets such as polytopes, which is our key objective.
>
> Hopefully, this difference in orientation explains why we omitted discussion of Mahendran & Vedaldi 2015 -- the extensive discussion of image based regularization and visualizations of reconstruction error are quite incomparable to what we are doing. That said, given that it's caused confusion, I'll clarify the relationship of our work to theirs. I was not familiar with Feather et al 2019, beforehand, but reviewing it now, I _believe_ that most of what I wrote about Mahendran & Vedaldi 2015 applies equally.
>
> The argument that max-pooling can be synthesized from relu is basically that
>
> max(x, y) = (x + y + relu(x - y) + relu(y - x)) / 2.

---

> > ### Comment · AnonReviewer3 · 2020-11-24
> > **follow up**
> >
> > Thanks to the authors for clarifying many of my questions. Much of this discussion about invertible networks and previous work using optimization based approaches should be included in any updated version of the paper.
> >
> > I can follow the discussion about novelty due to past work focusing only on single logits, however I don't quite understand this point:
> >
> > >The point being that domain-experts such as policymakers or scientists cannot productively work with the preimage of a single logit to answer more abstract questions from their fields. Our goal is not to use the preimage to show deep learning researchers some fundamental insight about neural networks, rather but to propose a tool that enable users to better interoperate with their instance of a neural network.
> >
> > If a tool is good for providing some fundamental insight about a neural network, then it naively seems like that should be sufficient for policy makers to similarly probe the representations of the network.  How does this work provide more insight into the preimage than previous work using optimization based approaches or invertible networks, especially given the intractability for large networks?

---

> > > ### Author Response · Authors · 2020-11-24
> > > **Many problems from safety, ethics, jurisprudence, etc. require more than just representative examples.**
> > >
> > > Thanks for your reply!
> > >
> > > We think that there are some insights about neural networks that DNN researchers would find profound or interesting that most users could not really appreciate (for example because it requires some subtle mathematical concept).
> > >
> > > Our point with the quote above can be more simply stated as: DNN researchers probably do not find it interesting that relu-based DNNs can be written as piecewise linear functions with a lot of kinks, but if one actually does the calculation (and, as far as we are aware, no one had!), then any numerate person can interesting and useful analysis, like visualize sensitivities or introspect on counterfactuals. Importantly, they need not be an expert on deep learning.
> > >
> > > At its simplest, we envision an aeronautical engineer navigating different projections of five to ten dimensional space of inputs to ACASs to uncover strange behaviors that are easy to recognize visually, but difficult to formulate beforehand.
> > >
> > > This example speaks to your last question:
> > >
> > > Optimization-based methods or invertible networks again operate only on single logits. The preimage of single points cannot be composed into a complete piecewise-linear characterization of the network, which is the only way to answer some questions. Many problems from safety, ethics, jurisprudence, etc. require more than just representative examples.
> > >
> > > In my reply to AnonReviewer1 I describe such a problem concretely -- one that a laywer or regulator might plausibly ask -- which can seeminly only be comprehensively answered using the piecewise linear formulation (about implausible "islands" of one classification surrounded by another).
> > >
> > > So yes, computing the preimage of a class label is intractable for large networks, but (1) the analysis we describe is most useful and insightful for small networks, (2) preimages can answer some interesting questions that to the best of our knowledge cannot be answered otherwise, and (3) even computing a single verification is intractable at scale.

---

### Decision · Program_Chairs · 2021-01-07
**Final Decision**

**Decision:**

Reject

**Comment:**

Thank you for your submission to ICLR.  The reviewers and I unanimously felt, even after some of the clarifications provided, that while there was some interesting element to this work, ultimately there were substantial issues with both the presentation and content of the paper.  Specifically, the reviewers largely felt that the precise problem being solved was somewhat poorly defined, and the benefit of the proposed preimage technique wasn't always clear.  And while the ACAS system was a nice application, it seems to be difficult to quantify the real benefit of the proposed method in this setting (especially given that other techniques can similarly be used to verify NNs for this size problem).  The answer that this paper provides seems to be something along the lines of "ease of visual interpretation" of the pre-image conditions, but this needs to be quantified substantially more to be a compelling case.